


# Enabling ice sheet models to capture solid Earth feedback with ease and accuracy

Surendra Adhikari[1], Lambert Caron[1], Holly K. Han[1], Luc Houriez[2], Eric Larour[1], and Erik Ivins[1]

[1]Jet Propulsion Laboratory, California Institute of Technology, Pasadena, CA 91011, USA
[2]Department of Mechanical Engineering, Stanford University, Stanford, CA 94305, USA

**Correspondence:** Surendra Adhikari (surendra.adhikari@jpl.nasa.gov)

**Abstract.** There is a growing consensus on the critical role that solid Earth processes play in influencing marine ice sheet dynamics over centennial timescales. A large body of literature shows that the feedback mechanisms associated with the solid Earth's gravitational, rotational, and deformational (GRD) response to ice mass loss slow the progression of ice sheet instabilities. However, due to the limited availability of efficient coupled system models, the specific characteristics of the feedback mechanisms, their sensitivities to ice sheet processes, and their impacts on sea level projections remain largely unexplored.
This paper introduces a straightforward method that enables ice sheet models to address this limitation by capturing GRD effects with virtually no additional software development and computational costs. The proposed method utilizes precomputed Green's functions and convolves these with ice mass changes within the ice sheet model primarily via matrix multiplication. This approach straightforwardly and accurately retrieves induced geoid and bedrock topography fields and paves the way for efficient coupling between ice sheet dynamics and leading-order solid Earth processes.

## 1 Introduction

An improved understanding of ice sheet dynamics is essential for producing accurate global and regional sea level projections. To achieve this, sophisticated ice sheet models that capture higher-order processes and feedback mechanisms are necessary. The ice sheet modeling community has increasingly recognized the importance of dynamic interactions between ice sheets and solid Earth processes over decadal and longer timescales. As ice mass changes, it exerts spatio-temporal variation in pressure on the solid Earth's surface, changing bedrock topography, geoid field, and thus sea level. These alterations in topography and sea level significantly influence marine ice sheet dynamics. They do so by affecting the retrograde bed slope, reinforcing pinning points and bedrock ridges, altering ocean heat transport patterns due to the modified shape of sub-shelf cavities, and influencing gravitational driving stress and surface mass balance through surface elevation and slope changes (e.g., Gomez et al., 2012; Adhikari et al., 2014; Larour et al., 2019; Albrecht et al., 2024; Kreuzer et al., 2025).



Several established methods of varying complexity exist for capturing solid Earth feedback in ice sheet models. The most straightforward approach treats the solid Earth as a two-layer system consisting of an elastic lithosphere that overlies a relaxing mantle half-space (e.g., Meur and Huybrechts, 1996; Ivins and James, 1999; Adhikari et al., 2014). A more complex method involves modeling a self-gravitating, density-stratified, viscoelastic Earth with a radially symmetric structure that typically includes an elastic lithosphere, a multi-layered Maxwellian mantle, and an inviscid core (e.g., Peltier, 1974; Farrell and Clark, 1976). Generally used in glacial isostatic adjustment (GIA) studies, this class of models enables capturing the full gravitational, rotational, and deformational (GRD; Gregory et al., 2019) feedback to ice sheet dynamics (e.g., Gomez et al., 2012; Larour et al., 2019; Han et al., 2022). More comprehensive approaches, such as those involving three-dimensional Earth structure (e.g., Albrecht et al., 2024; Gomez et al., 2024) and sophisticated non-Maxwellian mantle rheology (Houriez et al., 2025), are currently being explored to enhance our understanding of feedback mechanisms.

All these models come with important caveats, especially in light of their suitability for high-resolution coupled simulations. Downsides may range from being too simplistic (e.g., half-space models) to involving complex numerical implementations and demanding significant computational costs (e.g., global GIA models). As a result, many ice sheet models, including those involved in the Ice Sheet Model Intercomparison Project (ISMIP), often omit the GRD feedback, possibly leading to biased sea level projections over centennial timescales. GIA emulators (Lin et al., 2023; Love et al., 2024) and more realistic regional models (Swierczek-Jereczek et al., 2024) are emerging as promising options that require smaller computational costs. However, these approaches still require adapting the module into an existing ice sheet modeling framework, necessitating a certain level of model development. Additionally, in some cases, model fidelity remains a subject of scrutiny.

This paper aims to enable ice sheet models to effectively capture solid Earth feedback at the regional scale without necessitating further model development or incurring significant computational costs. Our approach leverages an extensive library of precomputed solid Earth response signals to a unit surface load, generally referred to as Green's functions, for plausible radially-symmetric Maxwellian Earth structures (Adhikari and Caron, 2024). We recast the convolution equation involving time-variable Green's functions and ice load changes in matrix formalism. This allows for accurate retrieval of the evolving geoid and bedrock topography fields as ice sheet models march forward in time and enables a straightforward coupling between the ice sheet and solid Earth processes. An efficient coupling enhances our understanding of feedback mechanisms, facilitates sensitivity analysis, and allows uncertainty quantification based on extensive ensemble simulations. Ultimately, these improvements will lead to more reliable projections of ice sheet contribution to sea level change.

## 2 Proposed method

Assuming that atmospheric pressure variability does not induce meaningful solid Earth deformation on decadal and longer timescales, the surface loading problem in the present context only requires resolving the lateral mass transport between the ice sheet and the ocean. A conditional function describing the ice and ocean load on the solid Earth surface, termed the "loading





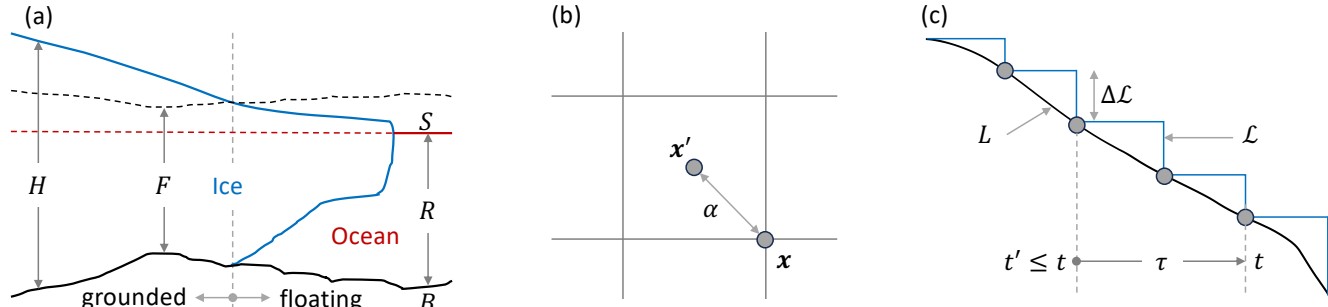

**Figure 1.** Schematics of the loading function and spatiotemporal grids. **(a)** Longitudinal cross-section of a marine ice sheet as it flows into the ocean. If ice thickness $H$ is larger than the flotation height $F$, a function of the bedrock topography $B$ and mean sea level $S$ (equation 2), it is assumed to be grounded, exerting pressure on the solid Earth's surface. Elsewhere in the ocean (including the floating ice shelves), ocean water (expressed here as the relative sea level $R \equiv S - B$) loads the underlying solid Earth (equation 1). **(b)** An example spatial grid illustrating the cell centroid $\boldsymbol{x}'$, where the elemental surface load is applied (equation 3), and the vertex $\boldsymbol{x}$, where the desired solid-Earth response signal is evaluated (equation 4). The two locations are separated by the great circle distance $\alpha$. **(c)** Temporal decomposition of the continuous surface load $L$ into discrete loading $\mathcal{L}$ using the Heaviside step function (equation 3). The discrete loading time $t'$ and the evaluation time $t$ are separated by $\tau$ (see equations 4 and 5).

function" $L$ [kg m$^{-2}$], at a given point in time $t$ may be expressed as follows:

$$L(\boldsymbol{x},t) = \begin{array}{ll} \rho_i H(\boldsymbol{x},t) & \text{if } H(\boldsymbol{x},t) > F(\boldsymbol{x},t) \\ \rho_w R(\boldsymbol{x},t) & \text{otherwise.} \end{array} \tag{1}$$

Here $H$ is the ice thickness [m], $R(\boldsymbol{x},t) = S(\boldsymbol{x},t) - B(\boldsymbol{x},t)$ is the mean sea level $S$ [m] relative to the bedrock topography $B$ [m], commonly referred to as the relative sea level, $\rho_i$ is the ice density [kg m$^{-3}$], $\rho_w$ is the ocean density [kg m$^{-3}$], and $\boldsymbol{x}$ is the 2-D position vector on the solid Earth surface. (Note that $S$ and $B$ must be defined relative to the same reference ellipsoid.) The flotation height for ice $F$ [m] follows the principle of hydrostatic equilibrium and is given by

$$F(\boldsymbol{x},t) = \frac{\rho_w}{\rho_i} \max\left[R(\boldsymbol{x},t),0\right]. \tag{2}$$

The condition in equation (1) implies that the solid Earth is loaded by the ice sheet in the grounded portion of the marine (and terrestrial) ice sheet and by the ocean otherwise (Figure 1 panel a).

Numerical modeling requires discretizing the continuous loading function into $p$ computational grids (having $q$ nodes) and $m$ time intervals: $[t_0,t_1]$, $[t_1,t_2]$, ..., $[t_{m-1},t_m]$. Spatial grids and time intervals do not have to be uniformly discretized. We may now express equation (1) – in the units of mass [kg] and with the new notation $\mathcal{L}$ – in its discrete form as follows:

$$\begin{aligned} \mathcal{L}(\boldsymbol{x}',t) &= \left[\bar{L}(\boldsymbol{x}',t_0) + \sum_{i=1}^{m}\left[\bar{L}(\boldsymbol{x}',t_i) - \bar{L}(\boldsymbol{x}',t_{i-1})\right]\mathcal{H}(t-t_i)\right]A(\boldsymbol{x}'), \\ &= \mathcal{L}(\boldsymbol{x}',t_0) + \sum_{i=1}^{m}\Delta\mathcal{L}(\boldsymbol{x}',t_i)\,\mathcal{H}(t-t_i), \end{aligned} \tag{3}$$

Here $\boldsymbol{x}'$ is the position of the center of the grid-cell or element, $A$ [m$^2$] is the elemental area, and $\bar{L}$ [kg m$^{-2}$] is the spatial mean of $L$ over that area. As shown in Figure 1 (panel b), we recommend placing the surface load at the grid centers, $\boldsymbol{x}'$, instead of at





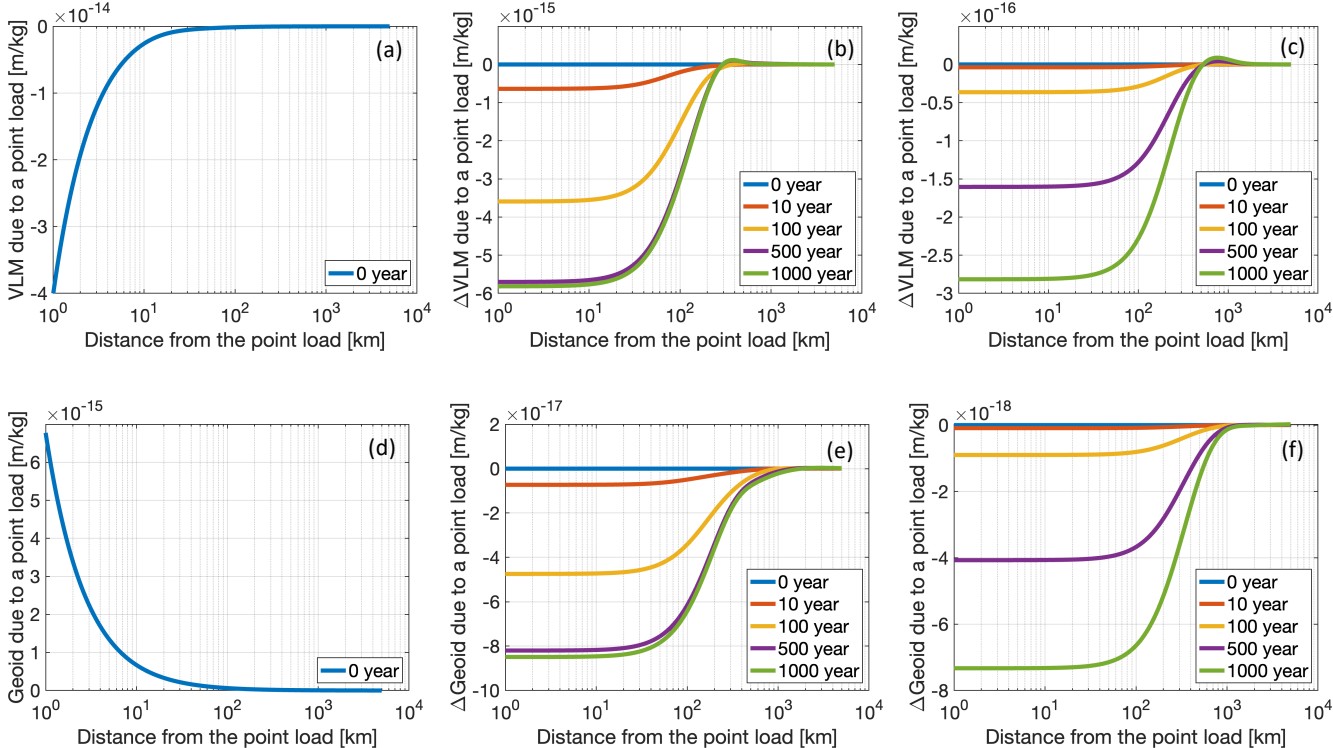

**Figure 2.** Time-dependent Green's functions for vertical land motion (VLM) (upper panels) and geoid (lower panels). Panels **(a)** and **(d)** show the elastic signals derived for the preliminary reference Earth model (Dziewonski and Anderson, 1981). Viscous signals, derived here by subtracting the elastic (panels a and d) from the total viscoelastic solutions, are shown for two earth models: **(b, e)** WAIS-earth and **(c, f)** EAIS-earth. WAIS-earth represents the laterally-averaged viscosity structure beneath the West Antarctica Ice Sheet (WAIS) featuring 50 km thick lithosphere and $1 \times 10^{19}$ Pa s upper mantle viscosity. EAIS-earth mimics the scenario beneath the East Antarctic Ice Sheet (EAIS), with 150 km thick lithosphere and $5 \times 10^{20}$ Pa s upper mantle viscosity. Lower mantle viscosity in both cases is fixed at $2 \times 10^{22}$ Pa s. Notice the contrasting response signals: solid Earth attains isostasy within ≈500 years in WAIS-earth, while in EAIS-earth, the noticeable viscous response starts emerging only after 100 years. Note the different y-axis limits and log scale on the x-axis.

the nodes or vertices, $\boldsymbol{x}$, where we assess the solid Earth response. This is important because the Green's functions we intend to use contain inherent singularities at the loading point. In the above equation, $\mathcal{H}$ is the Heaviside step function that takes the value of unity for $t \geq t_i$ and zero otherwise. It ensures that we impose the stepwise load change on the solid Earth surface at the end of each time interval (Figure 1 panel c). For the period $-\infty < t < t_1$, the surface load is held constant to its initial or reference value $\mathcal{L}(\boldsymbol{x}', t_0)$, and we assume that the solid Earth is in hydrostatic equilibrium. As such, only the change in load induces solid Earth response, which can be evaluated at any time $t \geq t_1$.

For $m = 1$ and in the limit of $p \to \infty$ for a structured mesh (or, equivalently, $A \to 0$), setting in equation (3) the otherwise zero loading function to unity strictly at a single computational grid and at time $t = t_m$ corresponds to a mathematical





description of a point load. Harmonic solid Earth response to the point load sustained on the surface of a radially symmetric layered solid Earth is traditionally computed as the time-dependent Love numbers (Love, 1909; Shida, 1912). We may assemble the Love numbers analytically to determine Green's functions (Appendix A). Adhikari and Caron (2024) calculated the time-dependent Love numbers for 483 radially stratified Maxwellian Earth models. They also derived corresponding Green's functions, relevant for estimating vertical land motion (VLM) and geoid change, which in the present context is equivalent to the mean sea level (MSL) change as defined in Gregory et al. (2019). These solutions are based on seismologically constrained elastic Earth structure (Dziewonski and Anderson, 1981) and include sufficient sampling of lithosphere thickness and upper mantle viscosity to investigate the seasonal to centennial timescale solid earth response to the evolution of ice sheets. Figure 2 shows Green's functions for two models, each representing the laterally-averaged Earth structure in West and East Antarctica based on seismic mapping and interpretations of Lloyd et al. (2020) and Ivins et al. (2023b).

Given the precomputed Green's functions (Adhikari and Caron, 2024), we can retrieve the solid Earth response signals induced by the loading function with any complexity through direct spatiotemporal convolution:

$$X(\boldsymbol{x},t) = G_X(\alpha,\tau) \otimes \Delta\mathcal{L}(\boldsymbol{x}',t'), \tag{4}$$

where $X$ is the GRD field of interest [m], $G_X$ is the corresponding Green's function [m kg$^{-1}$], $\alpha$ is the (great circle) distance between the loading position $\boldsymbol{x}'$ and where the response signal is determined $\boldsymbol{x}$, and $\tau$ is the time elapsed between the loading time $t'$ and when the response signal is evaluated $t$ (Figure 1). Let $\phi$ and $\lambda$ be the geographic latitude and longitude of the evaluation and loading points: $\boldsymbol{x} \equiv (\phi,\lambda)$ and $\boldsymbol{x}' \equiv (\phi',\lambda')$, respectively. Then, the distance between the two points can be written as $\alpha = r\left[2\arcsin\sqrt{\sin^2\left(\frac{|\phi-\phi'|}{2}\right) + \cos\phi\cos\phi'\sin^2\left(\frac{|\lambda-\lambda'|}{2}\right)}\right]$, where $r$ is the Earth's surface radius. Note that $\Delta\mathcal{L}$ in the above equation represents the change in load over the preceding interval to the loading time (Figure 1 panel c). In the present context, $X$ represents VLM or the bedrock topography $B$ and MSL or geoid $S$.

The convolution operator $\otimes$ appearing in equation (4) can be straightforwardly evaluated with scientific computational tools. For example, the desired GRD signal $X$ at a given point in space $\boldsymbol{x}_k$ and time $t$ and can be evaluated as follows:

$$X(\boldsymbol{x}_k,t) = \sum_{j=1}^{p}\sum_{i=1}^{m} G_X(\alpha_{k,j},t-t_i')\,\Delta\mathcal{L}(\boldsymbol{x}_j',t_i'), \quad \text{for } t_m' \leq t. \tag{5}$$

Once again, $p$ and $m$ are the total number of loading points in space and time. The GRD field can be obtained by iterating over discrete points $k \in [1,q]$, where $q$ is the total number of computational nodes (Figure 1 panel b). For simulations of ice/Earth coupling at a centennial timescale, we expect $q$ to be several orders of magnitude larger than $m$ (a direct function of coupling interval, typically 10 years). A more efficient convolution approach could involve retrieving the GRD field induced by a Heaviside load and iterating along the time dimension. Appendix B presents an algorithm for such a scheme.

## 3 Accuracy of the Proposed Method

We scrutinize the proposed method by comparing example GRD results against self-consistent solutions acquired from simulating the Ice-sheet and Sea-level System Model (ISSM; Adhikari et al., 2016; Larour et al., 2019; Houriez et al., 2025). In





the latter, self-consistency is sought in the ocean loading and GRD solutions for a given ice loading. We accomplish this by solving the sea level equation (Farrell and Clark, 1976; Milne and Mitrovica, 1998; Spada and Melini, 2019) on an unstructured Earth's surface mesh that conserves the total mass of ice and ocean. No regional model, including our proposed one, can
resolve self-consistency in global surface loading and the solid Earth response. The critical question is how well the regional model reproduces the self-consistent solutions. For identical representation of ice loading and solid Earth in the regional and global self-consistent models, the difference in predicted GRD solutions stems from the treatment of ocean loading and the rotational feedback. Here, we aim to quantify this difference for the predicted change in VLM and MSL. As such, our regional model solutions only include the ice loading effect (i.e., $R = 0$ in equation 1) and do not account for the rotational feedback,
which Larour et al. (2019) have shown to be negligible.

First, we examine the elastic Earth response to the observed trend in Antarctic ice mass change derived from the Gravity Recovery and Climate Experiment (GRACE) and its Follow On (FO) mission data. A key aspect of the load model is the significant mass loss occurring in the Amundsen Sea Sector and Wilkes Land, contrasted by a moderate mass gain in Dronning Maud Land (Figure 3 panel a). This pattern is also evident in the near field of self-consistent ocean loading (Figure 3 panel
b), where there is a reduction in ocean load near the areas of ice loss and an increase in ocean load near the areas of ice gain. Predicted VLM and MSL fields show minor differences between the regional and global models, implying the high accuracy of the proposed method. Self-consistent VLM solution is systematically larger (Figure 3 panel e) due to the reinforcing effect of the ocean load, but this is insignificant (Figure 3 panel f). An equally insignificant difference in the MSL field appears to be primarily dominated by the Earth's rotational effect (Figure 3 panels h and i).

Next, we evaluate the proposed method for two viscoelastic Earth models using a sample of the future evolution of the Antarctic Ice Sheet until 2300. We use precomputed ice sheet model solutions – 'experiment 05' of ISMIP6 Antarctica 2300 projections under the SSP5-8.5 scenario based on the UKESM climate model (Seroussi et al., 2024; Han et al., 2025) – and load the solid Earth every 10 years between 2010 and 2300 CE. In this load model, the ice sheet mass change does not contribute much to sea level change during the first half of the analysis period (Figure 4 panel a). The sea-level contribution
increases rapidly after around 2150, reaching a total change of about 1.5 meters. The cumulative ice thickness change suggests substantial mass loss from West Antarctica and along the coastal regions of East Antarctica (Figure 4 panel b). In contrast, the inland areas of the ice sheet achieve a modest mass gain. As introduced in Figure 2, we consider two Earth models representing the laterally-averaged structure beneath West and East Antarctica. In the case of WAIS-earth, the predicted responses of the solid Earth exhibit significantly larger amplitudes and more pronounced high wave number features (Figure 4 panel c) than
those for EAIS-earth (Figure 4 panel e). However, the predicted solutions derived from the regional and global models are very similar, irrespective of the choice of solid Earth model. These solutions differ only by a few percent in areas experiencing significant mass loss and subsequent bed uplift (Figure 4 panels d and f). Like in the previous experiment (Figure 3), regional and global model differences are systematic: the ocean load generally enhances the predicted VLM fields, while the rotational feedback dominates the disparity in MSL fields that show a contrasting pattern in East and West Antarctica.

Last, we examine coupled ice/Earth simulations for Thwaites Glacier, building on a recent study by Houriez et al. (2025), to assess the utility of our proposed method in simulating ice sheet dynamics. Their approach uses an anisotropic mesh to







**Figure 3.** Method accuracy for elastic Earth models. We force the solid Earth by the observed ice load, $dH/dt$, derived from the GRACE/-FO mission data over 2002-2024 (**a**). We quantify the associated self-consistent ocean load, $dR/dt$, by solving the sea level equation (**b-c**). We use the self-consistent solutions (not shown) for the bedrock topography and geoid change – that capture both the ice and global self-consistent ocean loads (panels a-c) – to validate the proposed method. (**d**) Our estimate of vertical land motion (VLM) rate, $dB/dt$, induced by the ice load alone (panel a). (**e**) The difference in VLM rate between the self-consistent solution and the regional solution over the Antarctic domain (shown in panel d). It is at least an order of magnitude smaller than the signal, especially in regions with larger displacement. Indeed, the two solutions appear virtually the same within the ice sheet domain (**f**), confirming the validity of the proposed method. (**g-i**) Same as panels (d-f), but for the rate of change in mean sea level (MSL), $dS/dt$.



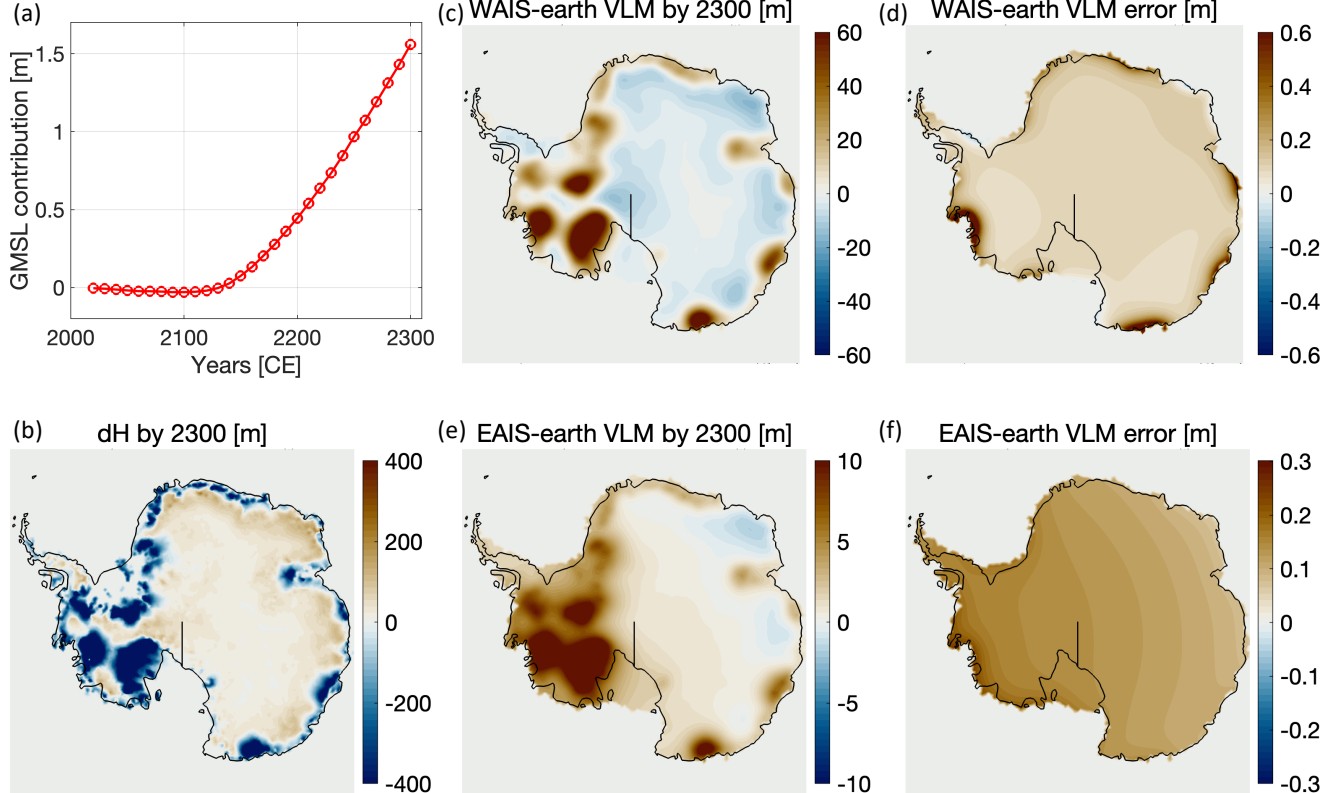

**Figure 4.** Method accuracy for viscoelastic Earth models. We use the evolving ice thickness data for 2010-2300 based on a solution that applied climate forcings from the recent Ice Sheet Model Intercomparison Project for CMIP6 (ISMIP6; Seroussi et al. (2024)) to a coupled ice-sheet/solid-Earth/sea-level model (Han et al., 2025): (**a**) Global mean sea level (GMSL) change at 10-year intervals between 2010 and 2300, and (**b**) the cumulative changes in (thickness equivalent) ice load, $dH$, over 290 years. (**c**) Total vertical land motion (VLM) by 2300 due to ice load alone, calculated for WAIS-earth using the proposed method. (**d**) Difference between the self-consistent solution (not shown) and the one shown in panel (c). (**e-f**) Same as panels (c-d), but for EAIS-earth. Regardless of Earth models, VLM errors are two orders of magnitude smaller than the signals, especially in regions with larger displacement.

capture kilometer-scale grounding line migration and allows for a coupling interval as short as one year. The ice sheet model is initialized by optimizing basal friction and ice rheology to match observed surface velocity (Rignot et al., 2014). It handles basal melt via PICOP parameterization (Pelle et al., 2019). Model inputs for ocean temperature, salinity, and surface mass balance are based on the Community Earth System Model (CESM) SSP5-8.5 scenario (Danabasoglu et al., 2020). The Earth model complies with the Maxwellian structure constrained by Global Navigation Satellite System (GNSS) data (Barletta et al., 2018) and further considers a transient relaxation of the asthenosphere and upper mantle based on the laboratory experiments of Extended Burgers materials (Faul and Jackson, 2015; Ivins et al., 2023a). Figure 5 shows predicted grounding line positions and sea level contributions from a standalone ice sheet simulation and two coupled simulations: one captures the ocean loading





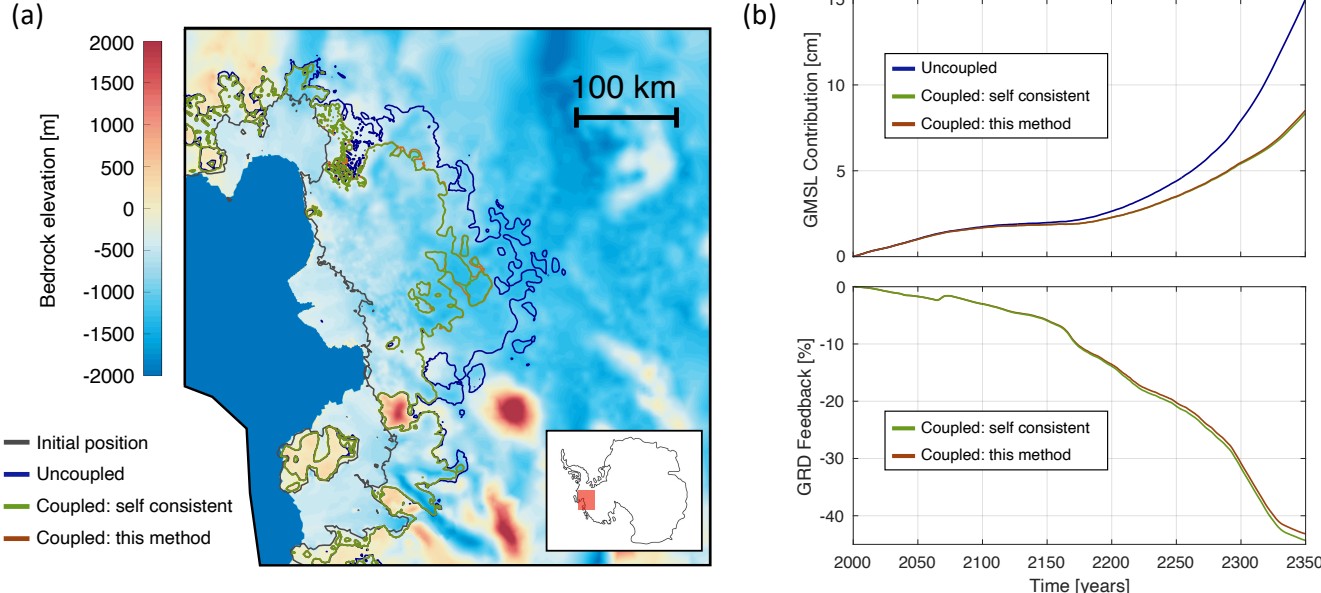

**Figure 5.** Method accuracy for coupled simulations. We leverage high-resolution coupled simulations for 2000-2350 that try to find consistency in ice sheet dynamics (hence, ice loads), Earth's gravitational, rotational, and deformational (GRD) response, and ocean mass distribution (Houriez et al., 2025). We reproduce key diagnostics of such simulations by coupling the ice sheet and solid Earth with the exclusion of ocean loads and rotational feedback. (**a**) Predicted grounding line positions at 2350 for standalone (uncoupled) ice sheet model and two coupled simulations. (**b**) Global mean sea level (GMSL) contributions and the GRD feedback captured in coupled simulations.

effect and rotational feedback (i.e., self-consistent solution), and the other does not (equivalently, the proposed method). The latter coupled model excellently reproduces the self-consistent solutions, capturing the GRD effect on sea level contributions within 1% for most of the time. This discrepancy increases up to 2% by the end of the simulation as a viscous response to ocean loading accumulates over time. However, it is insignificant compared to the large disparity in the centennial-timescale projections of the Antarctic Ice Sheet (Seroussi et al., 2024).

These examples demonstrate the high accuracy of the proposed method in capturing self-consistent solutions within the ice sheet domain. We show that the predicted MSL and VLM fields and their impact on ice dynamics are primarily influenced by the direct ice load, which the proposed method can easily handle. The impacts of ocean load and rotational feedback are minimal, although our method can still capture some of these signals. For instance, the ocean load near the ice sheet may be derived and refined iteratively from the VLM and MSL fields, as in the classical sea level solver. However, refined ocean loads

may not perfectly align with the self-consistent solutions. Additionally, the centrifugal potential perturbed by ice (and near-field ocean) and its effect on the MSL and VLM fields can be computed semi-analytically. Since the ice load alone captures most of the near-field signals with great accuracy – and given that our primary focus in this paper is on the straightforward retrieval of





the leading-order solid Earth signals within the ice sheet model domains – we do not recommend pursuing these higher-order signals, as it may require a more complex workflow for a minimal gain in solution accuracy.

**4 Conclusions**

We propose a straightforward method that allows ice sheet models to accurately capture solid Earth feedback on decadal to centennial timescales. This method only involves interpolating precomputed Green's functions and performing matrix multiplication with the evolving ice loading. Consequently, it does not introduce any numerical complexity or significantly increase the computational costs for ice sheet models.

We demonstrate the high accuracy of our method in capturing self-consistent solutions for the time-dependent geoid and bedrock topography fields within the ice sheet domain. Recent studies highlight the high sensitivity of solid Earth feedback to ice sheet model resolution and coupling time interval (Han et al., 2022; Houriez et al., 2025). In coupled simulations involving a global sea level solver, frequently capturing kilometer-scale features, such as subtle migrations of grounding lines and evolving bedrock ridges, can be exceptionally challenging. In contrast, our approach allows ice sheet models to determine

the spatiotemporal resolution of solid-Earth response signals without added complexity or significant computational costs. Therefore, the proposed method offers a strategic advantage over global self-consistent sea level models for ice sheet modelers seeking to capture solid Earth feedback in their simulations on centennial timescales.

The core idea of our method is to leverage precomputed Green's functions. An extensive collection of these functions is available for radially symmetric Maxwellian Earth models (Adhikari and Caron, 2024). This library includes many plausible

solid-Earth structures, from those with a thin lithosphere and a relatively softer upper mantle, such as that mapped in West Antarctica (Lloyd et al., 2020) or in Wilkes and Aurora Subglacial Basins in East Antarctica (Hansen and Emry, 2025), to models with a cratonic lithosphere and a stiffer mantle typically used in glacial isostatic adjustment studies. We aim to enhance this library by performing additional computations for more refined Earth structures and comprehensive linear viscoelasticity featuring transient rheology. Such data may be critical to accurately capturing the ice/Earth feedback at basin scales.

A primary motivation for this research is to enhance current and future undertakings of the Ice Sheet Model Intercomparison Project (ISMIP) by enabling greater participation in coupled ice/Earth simulations, with direct implications for future Intergovernmental Panel on Climate Change (IPCC) reports that cater to planners and policymakers. We, therefore, focus on capturing near-field signals over centennial and shorter timescales. The proposed method does not substitute a sea level solver, which is essential for global loading studies across timescales. This includes modeling glacial isostatic adjustment processes,

interhemispheric ice sheet interactions (Gomez et al., 2020), and far-field geodetic observables induced by ice sheets.

*Code availability.* In Appendix B, we provide an algorithm with some useful Matlab code blocks to facilitate the implementation of the proposed method. Our self-consistent model simulations were conducted using the Ice-sheet and Sea-level System Model (ISSM), an open-access software code available at https://github.com/ISSMteam.



*Data availability.* We provide time-dependent Maxwellian Love numbers and Green's functions for 483 combinations of lithosphere thick-
195 ness and upper mantle viscosity in Adhikari and Caron (2024). The lower mantle viscosity is fixed at $2 \times 10^{22}$ Pa s.

## Appendix A: Love numbers and Green's functions

The gravitational and deformational response of the solid Earth to a point load applied at its surface is commonly known as
the loading Love numbers. These Love numbers play a crucial role in loading studies, including glacial isostatic adjustment
theory, under the assumption of radial Earth symmetry. They are derived from the so-called $y_i$ system of equations based on
the principles of mass conservation, momentum conservation, and Poisson's equation (Alterman et al., 1959; Peltier, 1974).
Here, we leverage the recently coded Love number capability (Caron et al., 2025) of the Ice-sheet and Sea-level System Model
(ISSM) that solves the $y_i$ system in the Laplace domain for a suite of linear viscoelastic rheologies, including compressible
elastic, Maxwell, Burgers, and Extended Burgers materials. It employs the Post-Widder method to convert the spectral solutions
to the time domain (Spada and Boschi, 2006). The code has been optimized for parallel performance at high spherical harmonic
degrees, targeting to resolve kilometer-scale processes critical for understanding ice sheet and solid Earth interactions (Larour
et al., 2019; Houriez et al., 2025). It has been validated against community standards (Spada et al., 2011). The Love numbers
related to radial deformation, $h_n(t)$, and gravitational potential, $k_n(t)$, are particularly relevant here. Figure A1 shows example
Love numbers for two models representing the solid Earth beneath West and East Antarctica (also see Figure 2).

Time dependent Love numbers can be assembled to derive the displacement and geoid response to the surface point load
(Longman, 1962; Farrell, 1972). These response functions, called Green's functions, may be written as follows:

$$\mathcal{G}_U(\alpha, t) = D \sum_{n=0}^{\infty} h_n(t)\mathcal{P}_n(\cos\alpha), \tag{A1}$$

$$\mathcal{G}_\Phi(\alpha, t) = D \sum_{n=0}^{\infty} [1 + k_n(t)]\,\mathcal{P}_n(\cos\alpha). \tag{A2}$$

Here $\mathcal{G}_U$ and $\mathcal{G}_\Phi$ are Green's functions for VLM and geoid (or MSL), $\mathcal{P}_n$ are Legendre polynomials of degree $n$, $\alpha$ is the arc
length between the location of point load and the location at which the solid Earth response is evaluated (see Section 2), and
215 $D = 3/(4\pi r^2 \rho_e)$ is the dimensioning constant with $\rho_e$ denoting the mean Earth density.

We may evaluate the infinite sum in the above equations by truncating the series at a sufficiently high degree, say at $n = N$.
A truncation at $N = 10^4$ may be sufficient to capture high wave-number features, such as subtle migration of grounding lines or
uplift of subglacial mountains and ridges, critical for understanding the ice/Earth feedback mechanisms (Houriez et al., 2025).
However, it appears insufficient to yield accurate and smooth Green's functions, especially in the near field of the loading
point (Figure A2). Noting the asymptotic nature of $h_n(t) \to h_\infty$ and $nk_n(t) \to k_\infty$ as $n \to \infty$ (see Figure A1), Farrell (1972)
suggests employing the so-called Kummer's transformation to get rid of these noises. For a sufficiently high-degree truncation,







**Figure A1.** Viscoelastic Love numbers. We show example solutions for two Earth models introduced in Figure 2: WAIS-earth (left panels) and EAIS-earth (right panels). Love numbers $h_n$ (upper panels) and $k_n$ (lower panels) are relevant for vertical land motion (VLM) and geoid estimation, respectively. Note the asymptotic nature of $h_n(t)$ and $nk_n(t)$ towards their respective constants $h_\infty$ and $k_\infty$ as $n \to \infty$. These constants solely depend on the elastic structure of the Earth. For the preliminary reference Earth model (Dziewonski and Anderson, 1981), we find $h_\infty \approx h_{10,000} = -6.214$ and $k_\infty \approx 10^4 k_{10,000} = -3.055$. Notice the different y-axis limits, highlighting the contrasting response signals for the two Earth models. The low viscosity regime in WAIS-earth yields larger-amplitude viscous signals. The thinner lithosphere in this model implies a significant viscous response at high wave numbers ($n \approx 400$ as opposed to $n \approx 200$ in EAIS-earth).

we may invoke $h_\infty \approx h_N$ and $k_\infty \approx Nk_N$ and express Green's functions as follows:

$$\mathcal{G}_U(\alpha,t) \approx D\left\{ \frac{h_\infty}{2\sin(\alpha/2)} + \sum_{n=0}^{N}[h_n(t) - h_\infty]\mathcal{P}_n(\cos\alpha)\right\}, \tag{A3}$$

$$\mathcal{G}_\Phi(\alpha,t) \approx D\left\{ \frac{1}{2\sin(\alpha/2)} - k_\infty \ln\left[2\sin(\frac{\alpha}{2})\right] + \sum_{n=1}^{N}[k_n(t) - \frac{k_\infty}{n}]\mathcal{P}_n(\cos\alpha)\right\}. \tag{A4}$$





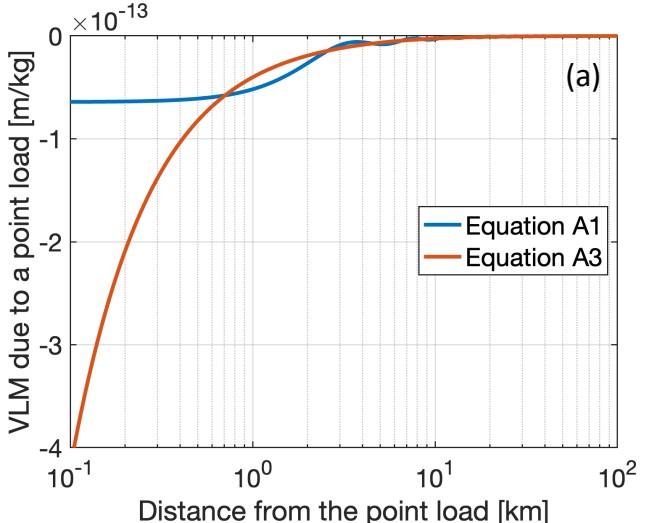
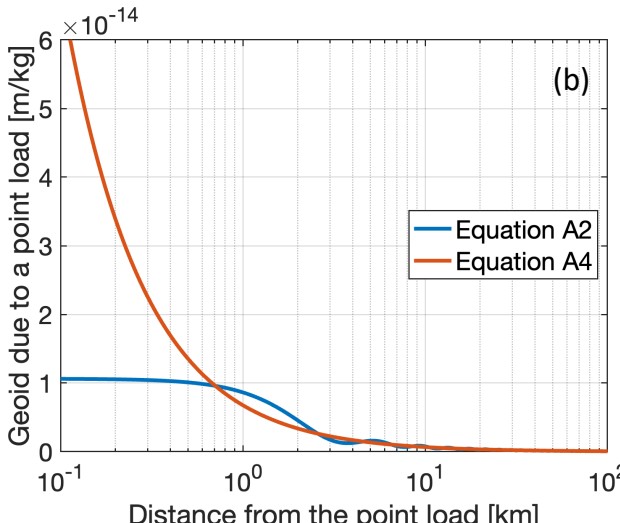

**Figure A2.** Evaluation of infinite sum in Green's functions. Since evaluating Love numbers up to infinitely large degrees is impractical, we have to truncate the sum appearing in equations A1 and A2 at a sufficiently high degree $n = N$. Even if we truncate the sum at $N = 10^4$, Green's functions oscillate in the near field of the applied point load (see blue lines). We handle these oscillations inherent to infinite sums by leveraging the asymptotic nature of Love numbers (see Figure A1) and using the Kummer transformation that partitions Green's functions into an analytic part whose exact solution exists and the finite sum that yields zero for $n \geq N$. The transformed expressions (equations A3 and A4) accurately capture Green's functions in the near field, hitting the inherent singularity at the loading point (red lines).

As shown in Figure A2, these expressions behave smoothly and are free from oscillatory artifacts. Adhikari and Caron (2024) deliver such solutions of Green's functions and corresponding Love numbers for numerous Earth models. These datasets resolve viscoelastic signals within $5,000$ km and over $1,000$ years after applying a sustained point load on the Maxwellian Earth surface. The spatiotemporal domain and sampling of these signals for a broad range of radially symmetric Earth structures will enable high-resolution coupled ice/Earth simulations on centennial timescales.

**Appendix B: A strategy for implementing the proposed method**

Here, we summarize a strategy that may be adapted to evaluate key matrices and convolution. The example we provide is written for Matlab and shall be adapted in any language.

**Distance matrix $\alpha(\boldsymbol{x}, \boldsymbol{x}')$:** We assume that the ice sheet mesh does not evolve in lateral dimensions, allowing us to compute the distance between the loading points (assumed to be elemental centroids) and evaluation points (assumed to be elemental vertices) only once. Let $\boldsymbol{x}_k \equiv (\phi_k, \lambda_k)$ for $k = [1, q]$ and $\boldsymbol{x}'_j \equiv (\phi'_j, \lambda'_j)$ for $j \in [1, p]$ be the geographic coordinates (in radians)
of elemental vertices and centroids. The distance matrix will be of $q \times p$ size, where $q$ and $p$ are the total numbers of evaluation and loading points in space (Figure 1 panel b). We may compute $\alpha$ as follows:





```
1:  for k = 1:q
2:      dphi = abs(phi2-phi1(k));
3:      dlam = abs(lambda2-lambda1(k));
4:      alpha(k,:) = r*2*asin(sqrt(sin(dphi/2).^2+cos(phi1(k)).*cos(phi2).*sin(dlam/2).^2));
5:  end
```

**Green's function $G_X(\alpha, \tau)$:** We assume that the loading time $t'$ and the evaluation time $t$ are relative to the same reference point and that their discrete representations are known *apriori*. It allows us to predetermine a set of unique non-negative $\tau$, where $\tau = t - t'$ (Figure 1 panel c), and prepare Green's function matrix only once before model run.

```
1:  tau = unique(transpose(t) - t_prime);
2:  tau = tau(tau >= 0);
```

We may now load "greensfunctions.nc" (Adhikari and Caron, 2024) and extract desired solutions for the chosen solid Earth model. All non-defined variables in the following example code are the default variables in the NetCDF file. (We may need to interpolate the solutions further if the desired lithosphere thickness or mantle viscosity does not match those provided.) In this example, we only load Green's functions required for VLM estimation.

```
1:  litho_idx = find(litho_thick==this_litho); % this_litho: user defined litho thickness [km]
2:  mant_idx  = find(mant_visco==this_mant);   % this_mant: user defined mantle viscosity [Pa s]
3:  greens_h_for_desired_earth = squeeze(greens_function_h(litho_idx,mant_idx,:,:));
4:  greens_h_at_desired_times = zeros(length(dist),length(tau));
5:  for ii = 1:length(dist)
6:      greens_input = greens_h_for_desired_earth(:,ii);
7:      greens_h_at_desired_times(ii,:) = interp1(eval_times,greens_input,tau);
8:  end
```

Finally, we may prepare a three-dimensional Green's function matrix, which will be untouched throughout the coupled model simulations. The following example uses a structure with dynamic variables to access specific Green's functions during convolution easily. The structure "Gvlm" has the same number of fields as the number of $\tau$ and each field has a size of $q \times p$, where $p$ and $q$ are once again the total number of elements and vertices.

```
1:  for ii = 1:length(tau)
2:      time_tag = ['tau ',num2str(tau(ii))];
3:      Gvlm.(time_tag) = interp1(dist,greens_h_at_desired_times(:,ii),alpha);
4:  end
```



**Loading function** $\Delta\mathcal{L}(\boldsymbol{x}',t')$: Let $m$ be the total number of Heaviside loads prior to the specific coupling time $t$ (Figure 1 panel c). The loading function $\Delta\mathcal{L}(\boldsymbol{x}',t')$ would have a size of $p \times m$, where $p$ is the number of elements. The ice sheet model computes this function (in units of kg), which we refer to in the following code as "deltaload".

**Convolution** $X(\boldsymbol{x},t) = G_X(\alpha,\tau) \otimes \Delta\mathcal{L}(\boldsymbol{x}',t')$: We may perform the spatiotemporal convolution and retrieve the desired

GRD signal (vertical land motion "vlm," in this example) as follows.

```
1:  vlm = zeros(q,1);
2:  for ii = 1:m
3:      tau = t - t_prime(ii);
4:      time_tag = ['tau',num2str(tau)];
5:      vlm = vlm + sum(bsxfun(@times,Gvlm.(time_tag),transpose(deltaload(:,ii))),2);
6:  end
```

*Author contributions.* SA conceived and carried out the research and wrote the first draft of the manuscript. LC and EI helped compute Love

numbers. HH contributed to validating the proposed method. LH and EL performed coupled simulations. All authors reviewed and approved the manuscript.

*Competing interests.* The authors declare no competing interests.

*Acknowledgements.* This research was conducted at the Jet Propulsion Laboratory, California Institute of Technology, under a contract with the National Aeronautics and Space Administration (NASA). Funding support was provided by NASA's Sea-level Change Team (N-SLCT),

Earth Surface and Interior (ESI) Focus Area, the Modeling, Analysis, and Prediction (MAP) Program, and the Cryosphere Sciences Program. This work was inspired by discussions within the ISMIP7 GIA/Sea-level Focus Group.



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
