# Peer review of "Enabling ice sheet models to capture solid Earth feedback with ease and accuracy"

_EGUsphere, 2025_

## Referee Comment (RC1)

**Review of Adhikari et al. 2025: Enabling ice sheet models to capture solid Earth feedback with ease and accuracy**

The authors propose in this manuscript a new method to consider the viscoelastic response of a viscoelastic earth to surface loading. The method is designed to consider the solid-earth feedback mechanism in coupled dynamic ice sheet models. In order to reach an efficient set up, they made some simplifications with respect to the usual solution of the glacial isostatic adjustment problem, here called GRD.

The main advantage I see in the consideration of pre-calculated Green's functions. They suggest an efficient algorithm for the convolution, although I would argue that such an algorithm should be state of the art. Furthermore, the main aspects of the algorithm only appear in the Appendices.

Only after reading it becomes clear that this algorithm is not sufficient to analyse the SE-IS feedbacks for longer time scales, when efficiency breaks down and also the accuracy is reduced. Furthermore, the sea level equation is not considered regarding mass conservation, ocean loading and also the rotational feedback are not considered in this approximation.

Whereas the authors rise the impression that the GRD response is considered, at the end their approach turns out to solve the sea level equation in its simplest form (Spada et al., 2019), i.e., without mass conservation, ocean loading and rotational feedback, providing the local relative sea level change due to radial displacement and gravity change from the ice-mass change only. They should set this premise at the beginning of the study and motivate it from the specific application to solid earth--ice sheet model coupling on decadal to centennial time scales.

In particular, they show that these simplifications are sufficient for modelling the future evolution of the West Antarctic ice sheet during the coming centuries giving an error of up to a few %.

I see this approach valid for the applications described, but I question its help when working over longer time scales, like a glacial cycle, when interpreting geodetic observables, or when considering lateral variations in earth structure consistently. In so far, I see this model as a small step towards a more realistic coupling of the solid earth to ice sheet dynamics, but not sufficient in order to achieve a physically consistent solution.

Mz summary is minor, as the presentation of the results can be be improved. Furthermore I recommend to transfer this manuscript to GMD, which I see more on the side of discussing software developments (the reason why I set novelty to Fair). In particular for TC, I do not see a major impact, also due to the fact that the applicability of this method was discussed only for the future evolution of West Antarctica. Otherwise, the setup is ok, and my further recommendations remain valid also after a possible transfer.

**Further recommendations:**

- 1. L. 7: I would skip 'virtually additional software development'. My interpretation is, that only an offline coupling is necessary, where only the interface has to be set. But such approaches were already applied in a number of coupled models like PISM (Albrecht et al., 2024), and MPI-ESM (Mikolajewicz et al., 2025 <a href="https://doi.org/10.5194/cp-21-719-2025">https://doi.org/10.5194/cp-21-719-2025</a>). Only to list those in which I was involved.
- 2. In the Introduction I miss a more structured discussion to which time and spatial scale the method is

- designed for.
- 3. L. 22ff: Regarding the solid earth response, I miss the discussion of ELRA and Lingle and Clark, which are both considered for instance In PISM, with more realistic viscoelastic displacements but no gravity change.
- 4. L. 28: I miss here the reference of Konrad et al., (2015, <a href="http://dx.doi.org/10.1016/j.epsl.2015.10.008">http://dx.doi.org/10.1016/j.epsl.2015.10.008</a>).
- 5. L. 36: Konrad et al. (2016, <a href="https://dx.do.org/10.2312/polfor.2016.005">https://dx.do.org/10.2312/polfor.2016.005</a>), compared ELRA against a global GIA model representing the solid earth response in a coupled system.
- 6. L. 38: Coming back to my 1. suggestion.
- 7. I don't see the requirement of adapting a GIA module in an existing code. Regarding VILMA it was done for CLIMBER-X (Willeit et al. 2024, <a href="https://doi.org/10.5194/cp-20-597-2024">https://doi.org/10.5194/cp-20-597-2024</a>) but for PISM and MPI-ESM it was done off line, by only exchanging the RSL and Ice thickness.
- 8. L. 55: Also, I would reserve an own equation for the definition of R.
- 9. L. 74: p was not introduced.
- 10. L 82ff: The authors should specify the sampling and set up of the viscosity structures. Furthermore, they state that the time range extend from seasonal to hundreds of years. So, why do they not consider anelastic effects in their Green's functions in this study?
- 11. L. 99ff: The discussion regarding the number of integrations comes a bit late, and results in a further problem which weakens the statement of 'ease' a bit. Can the author give realistic integration times? That would help the reader to estimate which further approximations have to be made.
- 12. L. 105: I would expect 'validate' instead of 'scrutinize' the proposed method.
- 13. L. 113ff: I wonder why the authors do not consider the floating condition here. And is the rotational feedback only neglected in this experiment or everywhere. This is only clarified in the last paragraph of this study.
- 14. L. 121: I would not translate 'minor differences' into 'high accuracy', but into 'sufficient accuracy'.
- 15. L. 123: the neglect of ocean load is only insignificant when considering small changes in sea level.
- 16. Figure 3: I miss the error in R as this is the main quantity when coupling is considered.
- 17. L144: Although it becomes clear from the reference, a short note what PICOP means would help.
- 18. L 147: A transient rheology is considered in the fully coupled setup. Does this mean this is also considered for the Green's functions applied for the simpler model, and so is also considered in the provided set of Green's functions, or alternatively additional transient effects are not necessary for the coupling.
- 19. L 153: They state that for the year 2350 AD a cumulated error bias only reaches 2%, for their experiment which is rather small. But may it not affect the timing of tipping points. Also, what about local effects, which are not represented in discussing the GMSL.
- 20. In Figure 5: I miss a comparison with LC or ELRA which are further coupled set ups.
- 21. L 152ff: For the process I agree, that the effect is small. What about the impact on far-field aspects? Do we need a different model approach then? So, my impression is that the rotational potential is not considered and so, the degree 21 pattern in the far-field sea level is not modelled correctly. Furthermore, the ocean loading is not considered. Which results in an additional difference.
- 22. L 163ff: I read this between the lines: This approach does not allow a straight forward representation

- of a self-consistent GRD. Additional effort would be necessary to reach this, but for the considered case of analyzing centennial coupling and near field processes it is sufficient.
- 23. L. 166ff: Here it becomes more clearly, where the method is applicable. Nevertheless, I wonder regarding 'accurately' and 'high accurate'. That the method holds only when coupling on decadal to centennial, I agree. At L. 83 you state also 'seasonal', so be consistent.
- 24. L. 170: You capture the GRD by a number of simplifications, which you should summarize here in view of what terms do not have to be considered in the intended scenarios.
- 25. L. 176: Why does the neglection of a number of processes result in a strategic advantage. Migration of grounding lines, and bedrock ridges, are rather important in the discussion of grounding line evolution, especially when reaching spatial resolutions of the ice sheet models.
- 26. L. 178: Love numbers of radially symmetric viscoelastic earth models are also accessible with public domain software like TABOO. Can such sets be implemented in your setup?
- 27. L 182: Is the extension to more comprehensive linear rheologies necessary regarding the number of simplifications you apply and also the statement on L. 154, that the largest uncertainties are in the ice-sheet models themselves?
- 28. L.185ff: This final statement appears rather late to my understanding. It reads rather like a premise for the development of the code. So, I would expect this motivation and also the disclaimer at the beginning of the manuscript. This would make the discussion of the study much more transparent.

Volker Klemann

---

## Referee Comment (RC2)

Adhikari et al. provide a method for rapidly calculating the gravitational and deformational solid Earth response (glacial isostatic adjustment) to ice sheet loading/unloading that is applicable for use near-field of ice sheets on decadal to centennial timescales. The approach outlined in this paper will be useful for ice sheet modelers who are interested in including accurate gravitational and deformational solid Earth feedbacks with low computational cost. This is a beautifully written paper that clearly articulates the method proposed, which is a simplification of a global gravitationally self-consistent sea level model.

Essentially, the authors make the argument that in places like Antarctica, the gravitational and deformational response to the ice load is the dominant control on the solid Earth response, and it is reasonable to ignore far-field ice sheet changes, ocean loads, or true polar wander. If this is the case, then calculating the gravitational and deformational response simply requires a single convolution calculation (convolving precomputed Green's functions with a given load history). They show that viscoelastic calculations using this approach differ by only a few percent compared to global self-consistent sea level models, and this small difference is dominated by rotation. Such a simplified approach can easily be adopted by ice sheet modelers to include gravitational and deformational feedbacks in dynamic ice sheet simulations and provides a computationally effective method for high resolution simulations. The authors have published a previous paper that provides a large suite of precomputed Green's functions for different Earth models, so ice sheet modelers have a range of 1D Earth structures to select from, and may even be able to better represent uncertainty on Earth structure by performing simulations with many different 1D Earth models.

There are limits to the proposed approach: it is not appropriate for thinking about global feedbacks or long-term stability of ice sheets in response to glacial isostatic adjustment. We would guess that on the millennial timescale (modeling for more than 1000 years), it will be important to include global ice sheets (global sea level change and far field effects), ocean loads, and rotation in order to match the global self-consistent sea level model. This approach also ignores 3D Earth structure heterogeneity, an important point to mention for a paper focused on Antarctica, where lateral heterogeneities can significantly alter glacial isostatic adjustment predictions (e.g. Lucas et al, Cryosphere, 2025)

Below we list a few suggestions for strengthening the manuscript:

1) More specific context for readers on WAIS-earth structure:

We recommend providing additional information in the manuscript about regional variability in West Antarctic Earth structure. While the authors state in the paper that these two Earth models represent laterally averaged Earth structure in West and East Antarctica, respectively, it would be more accurate to say the Earth model selected to represent WAIS specifically represents coastal West Antarctica, specifically the Amundsen Sea Embayment sector, and not laterally averaged Earth structure across the entire WAIS. The authors justify their choice of both WAIS-earth and EAIS-earth models by referencing the Lloyd et al. (2020) and Ivins et al. (2023b) studies in Lines 84-84; however, neither of these studies suggest an average lithospheric thickness of 50 km across West Antarctica. Lloyd et al. (2020) does not quantitatively discuss lithospheric thickness in Antarctica. Instead, we suggest referring to the lithospheric thickness estimates from Wiens et al. (2023) and Brown & Fischer et al. (2025). These studies find 60-100 km lithosphere across West Antarctica, suggesting the choice of a 50 km lithosphere as a laterally-averaged value is not consistent with seismic constraints for West Antarctica.

We are happy with the choice to include the Earth model selected to compare against the EAIS-earth model (we don't see a need to change all of the figures or results), as long as the authors specify that this Earth model represents an end member rather than an average value for the WAIS, and that this earth structure is likely representative of specific regions (e.g. Amundsen Sea Embayment). We suggest citing studies that justify this Earth model for these specific regions in coastal West Antarctica, since such an Earth model (50 km lithosphere and 10^19 Pa s upper mantle) is certainly not appropriate for places like Weddell Sea or Ross Sea sectors. Explicitly explaining the choice of Earth model and the applicable region will be important for an audience of ice sheet modelers who need to decide which earth model is appropriate for their location of study.

Because significant ice sheet retreat is projected in the Weddell and Ross sectors over the next ~3 centuries, it will be useful to explain to readers that these are regions where there is lithosphere >50 km and upper mantle viscosities >10^19 Pa s. Accurately capturing the solid Earth and sea level response to ice mass loss in these sectors will therefore require an Earth model different from that adopted here.

Given these comments, we believe the manuscript would be strengthened by adding a paragraph that discusses the best choice of Earth models to use for different parts of Antarctica. Such additional text will be especially useful for a paper geared towards the ice sheet modeling community, as readers will want to know that there is not a single Earth models they should use to represent West Antarctica. These users should be aware that such a thin lithosphere and low-viscosity upper mantle represents one of the Earth structure end members in West Antarctica. We think this would broaden adoption of the proposed method to explain more clearly how ice sheet modelers can select from this large suite of 1D Earth models.

2) More discussion of the appropriate use of this methodology:

It would be helpful for the authors to include more discussion about the appropriate use (and limitations) of the proposed methods in the discussion/conclusions section. For example, even though the authors present two different Earth models for Antarctica, the proposed method ultimately requires a single 1-D Earth structure, which in not realistic across all of Antarctica. The authors need to mention this point to better situate their proposed method with respect to other methods. You might include reference to other efforts aiming to approximate bedrock deformation assuming laterally heterogeneity (i.e., van Calcar et al., in review: "Approximating ice sheet – bedrock interactions in Antarctic ice sheet projections"). It will also be useful to explicitly state the appropriate timescales and questions that this method targets. For example, adding a sentence or two that describes the potential uses of this methodology and also notes its limitations (multi millennial timescales etc.). While there is a sentence in the text that mentions long-term stability and global feedbacks, it should be explicitly stated somewhere what timescale limitations there might be using this method. Explain when it will be important to include rotation, ocean loads, and far-field effects. Describing the appropriate uses of this methodology will also be useful to emphasize in the abstract and introduction - and if the authors really wanted to look in detail they might provide information in the supplement showing at what timescale there is no longer a good fit between this methodology and that of a globally self-consistent model.

Line by line comments:

- Abstract: Need to specify somewhere in the abstract that the proposed approach assumes 1-D / laterally homogeneous Earth structure.
- Line 64: the dashes in this line are slightly confusing, may be good to phrase this without the dashes.
- Line 81: can you define mean sea level here rather than referencing a paper. MSL is referring to see surface height, is that correct? It is a little confusing to have GMSL and MSL in the same paper, it might be less confusing to use a different acronym.
- Similarly S is a confusing symbol for the geoid when G is usually used in papers on the sea level equation, we suggest changing Gx to GFx and making G the geoid variable, this would be simpler for the sea level community to follow given what past variable names have been used for
- Rather than explaining the variables in the text, it would be useful to include a table with all the variable names as an easy reference for the reader
- Line 108: I think of Kendall et al. 2005 as a standard reference for the sea-level equation, might consider including
- Line 171: Consider adding additional references that explore coupling GIA and grounding line predictions at high resolution: Kodama, et al. "Impact of glacial isostatic adjustment on zones of potential grounding line stability in the Ross Sea Embayment (Antarctica) since the Last Glacial Maximum." *EGUsphere* 2024 (2024): 1-25., Wan, Jeannette Xiu Wen, et al. "Resolving glacial isostatic adjustment (GIA) in response to modern and future ice loss at marine grounding lines in West Antarctica." *The Cryosphere* 16.6 (2022): 2203-2223.

Figures

- Figure 1c: In the caption, it would be useful to specify that this figure represents an ice load that is shrinking through time to make the figure easier for the reader to interpret.
- Figure 2: why is unit in meter per kilogram?
- Given the importance of the WAIS-earth and EAIS-earth models to the entire manuscript, please move (or at least repeat) the description of the WAIS-earth and EAIS-earth models to the main text instead of putting this description in the caption of Figure 2.
- Figure 3c: We suggest showing a global map instead of just the northern hemisphere. It is somewhat confusing to only see a portion of the northern hemisphere. A global map would help the readers better understand what is happening in Antarctica.
- Figure 4: It would be informative to include cumulative mean sea level changes in Figure 4, similar to how figures of mean sea level are included alongside vertical land motion in Figure 3. Plots of mean sea level in Figure 4 would be relevant to the points made in the discussion section, comparing the self-consistent model to the proposed method.
- Method accuracy for viscoelastic Earth models is only compared for 2300 in Fig. 4. It would be useful to ice sheet modelers to see other time steps comparing method accuracy, i.e., 2150. Such a figure could be included in the supplement perhaps.